# Phenotype of the Aging-Dependent Spontaneous Onset of Hearing Loss in DBA/2 Mice

**DOI:** 10.3390/vetsci8030049

**Published:** 2021-03-17

**Authors:** Min-Soo Seo, Byeonghyeon Lee, Kyung-Ku Kang, Soo-Eun Sung, Joo-Hee Choi, Si-Joon Lee, Young-In Kim, Young-Suk Jung, Un-Kyung Kim, Kil Soo Kim

**Affiliations:** 1Laboratory Animal Center, Daegu-Gyeongbuk Medical Innovation Foundation, Daegu 41016, Korea; msseo@dgmif.re.kr (M.-S.S.); kangkk@dgmif.re.kr (K.-K.K.); sesung@dgmif.re.kr (S.-E.S.); cjh522@dgmif.re.kr (J.-H.C.); sjlee1013@dgmif.re.kr (S.-J.L.); 2Department of Biology, College of Natural Sciences, Kyungpook National University, Daegu 41016, Korea; lbh2080@knu.ac.kr; 3Advanced Bio-Resource Research Center, Kyungpook National University, Daegu 41566, Korea; 4KPC Corporation, Gwangju 12773, Korea; kimyoungin@kpclab.co.kr; 5College of Pharmacy, Pusan National University, Busan 46241, Korea; youngjung@pusan.ac.kr; 6KNU Creative BioResearch Group (BK21 Plus Project), School of Life Sciences, Kyungpook National University, Daegu 41016, Korea; 7Department of Veterinary Toxicology, College of Veterinary Medicine, Kyungpook National University, Daegu 41566, Korea

**Keywords:** hearing loss, laboratory animals, aging, DBA/2 mice

## Abstract

**Simple Summary:**

In this study, we confirmed the changes in hearing function and inner ear structure over a long period of time in DBA/2 mice, a laboratory animal model suitable for studying hearing loss. We believe that our study is the first to report findings regarding hearing function and structural changes in DBA/2 mice aged ≥34 weeks. These results are of significance for researchers and the scientific community using laboratory animal models.

**Abstract:**

DBA/2 mice are a well-known animal model for hearing loss developed due to intrinsic properties of these animals. However, results on the phenotype of hearing loss in DBA/2 mice have been mainly reported at an early stage in mice aged ≤7 weeks. Instead, the present study evaluated the hearing ability at 5, 13, and 34 weeks of age using DBA/2korl mice. Auditory brainstem response test was performed at 8–32 KHz at 5, 13, and 34 weeks of age, and hearing loss was confirmed to be induced in a time-dependent manner. In addition, histopathological evaluation at the same age confirmed the morphological damage of the cochlea. The findings presented herein are the results of the long-term observation of the phenotype of hearing loss in DBA/2 mice and can be useful in studies related to aging-dependent hearing loss.

## 1. Introduction

With the development of modern civilization, individuals are at an increased risk of exposure to excessive noise. Moreover, with the increase in the number of aging individuals, senile hearing loss is emerging as a major social and environmental concern [1]. As reported by the World Health Organization, hearing loss has increased worldwide, with 360 million individuals currently suffering from hearing loss [2]. It is known that senile hearing loss increases gradually at first but then increases rapidly with age [3].

Studies using laboratory animals have great merits in that the research environment can be controlled [4]. In addition to supplying drinking water and food; preventing exposure to the external environment; and controlling light, temperature, humidity, and noise; the age of the laboratory animal can be selected according to the study situation [5]. In this case, animal ages for examination of hearing functions were selected to reflect changes in hearing typical of senile hearing loss [6,7,8].

In this study, DBA/2 mice, which are frequently used in hearing-loss-related research studies [6,9], were used to measure the changes in hearing threshold with increasing age using the auditory brainstem response (ABR) test to evaluate the structural changes in the cochlea by histopathological analysis. Through these approaches, we aimed to obtain basic data for hearing loss research in DBA/2 mice.

## 2. Materials and Methods

### 2.1. Animals

Male DBA/2korl mice were kindly provided by the Department of Laboratory Animal Resources at the National Institute of Food and Drug Safety Evaluation (NIFDS, Cheongju, Korea). The mice had access to a standard irradiated chow diet (Purina, Seoul, Korea) and sterilized water ad libitum. In this study, the animal facility conditions were maintained under a strict light cycle (light on–off system, 12 h dark–light cycle, 07:00–19:00), 23 ± 2 °C, and 50 ± 10% relative humidity under specific pathogen-free conditions. To evaluate hearing loss in the DBA/2korl mice, we used 37 male mice for ABR test and 15 male mice for histopathological analysis. ABR was recorded and histopathological analysis was conducted at 5, 13, and 34 weeks of age. The animal experimental protocols were reviewed and approved by the Institutional Animal Care and Use Committee of The Laboratory Animal Center of the DGMIF (IACUC; approval No. DGMIF-20011601-01) and were in accordance with their guidelines.

### 2.2. ABR Test

Auditory functions were assessed by recording ABRs using an ABR workstation (System 3; Tucker Davis Technology (TDT), Inc., Alachua, FL, USA), as described previously [10]. All tests were conducted in a soundproof room. Briefly, the mice were anesthetized by an intramuscular injection of a mixture of tiletamine–zolazepam (1.8 mg/100 g) and xylazine hydrochloride (0.7 mg/100 g). The mice were then placed on a heating pad to maintain their body temperature at 37 °C. The body temperature of the mice was monitored using a rectal thermometer. ABRs were recorded by inserting subcutaneous needle electrodes into the vertex (+charge), mastoid (−e), and hind leg (ground). Acoustic stimuli, which included a tone-burst stimulus with a 1 ms rise/fall time and a 5 ms plateau at frequencies of 8, 16, and 32 kHz or transient click stimuli, were monaurally applied using a speaker. Stimulus signals were generated using SigGenRP and an RP2.1 real-time processor and were transmitted using a programmable attenuator (PA5 and TDT), a speaker driver (ED1 and TDT), and an electrostatic speaker (MF1 and TDT). The stimuli were generated for 500 repetitions at 5 dB decrements, starting from a sound pressure level of 90 dB to the acoustic threshold at each frequency. The phase of the stimulus was reversed after each presentation to decrease the artifacts caused by repetitive stimuli.

### 2.3. Histological Analysis

After recording the ABRs, the mice were perfused with 4% paraformaldehyde (PFA) in phosphate-buffered saline (PBS) and their inner ears were isolated. For preparing paraffin sections, the inner ears were fixed with 4% PFA in PBS for 24 h at 4 °C and then decalcified with 10% ethylenediaminetetraacetic acid in PBS for 24 h at 4 °C. The specimens were dehydrated using a graded ethanol series, permeabilized with xylene, and embedded in paraffin. The paraffin-embedded inner ears were serially sectioned to obtain 6-μm-thick sections using a microtome (Leica RM2235; Leica Microsystems, Nussloch, Germany), mounted on Superfrost Plus microscope slides (Fisher Scientific, Pittsburgh, PA, USA), and then subjected to H&E staining.

### 2.4. Statistical Analysis

The results were analyzed using Student’s *t*-test and Excel (Microsoft, Redmond, WA, USA) and are presented as mean ± standard error in graphical plots. Statistically significant differences are indicated using asterisks. *p* values of <0.0001 were considered statistically significant.

## 3. Results

### 3.1. Measurement of Auditory Function with Increasing Age

In the present study, four stimuli (transient click and 8, 16, and 32 kHz) were delivered at 5, 13, and 34 weeks of age in the mice, according to the average hearing thresholds (Figure 1A–D). At 5 weeks of age, a hearing threshold value of ≥40 dB was confirmed at a high-frequency stimulation of 32 kHz. From 13 weeks of age, the average hearing threshold value appeared to increase rapidly, thereby causing hearing loss. In particular, a hearing threshold value was recorded of ≥70 dB in transient click and 8 kHz stimuli and ≥80 dB in 16 and 32 kHz stimuli in a 34-week-old elderly mouse. These results indicate that significant hearing loss was developed at an early stage in the DBA/2korl mice, especially at high frequencies (Figure 1A–D). In addition, it was confirmed that changes in the hearing threshold were markedly increased in an aging-dependent manner.

### 3.2. Histopathological Analysis of Cochlea with Increasing Age

To observe the structural changes in the cochlea of the inner ear with increasing age in DBA/2korl mice, a histopathological analysis was performed using H&E staining at 5, 13, and 34 weeks of age (Figure 2A–C). The structures of the apex, middle, and basal parts of the cochlea were confirmed. At 5 weeks of age, a normal histopathological structure was observed (Figure 2A). At 13 weeks of age, degeneration of the hair cells in the middle turn and basal turn of the organ of Corti was observed (Figure 2B), which became particularly pronounced at 34 weeks of age, when overall degeneration was observed. Furthermore, it was confirmed that the spiral ganglion exhibited degeneration at 34 weeks of age (Figure 2C). These results suggest that hearing loss is associated with damage to the structure of the cochlea.

## 4. Conclusions

DBA/2 mice, an inbred mouse strain, have been widely used as disease models for research related to hearing loss and neuronal degeneration [9,11]. DBA/2 mice have three recessive alleles focused on Cdh23ahl, which can induce progressive cochlear damage affecting the inner ear’s organ of Corti [12]. DBA/2 mice have normal hearing function at 2–3 weeks of age, but then they gradually develop hearing loss in the high-frequency region. It has been reported that at 5 weeks of age, the hearing threshold increases by 30 dB at ≥8 kHz and severe hearing loss occurs at 2–3 months of age [13]. However, to the best of our knowledge, hearing function and histopathological findings have not been reported in DBA/2 mice aged ≥24 weeks to date. The ABR test is the primary noninvasive method used to evaluate hearing function, and its findings have been previously reported in laboratory animals [14,15,16]. Therefore, the aim of our study was to investigate hearing function and histopathological findings in elderly DBA/2 mice.

In the present study, the ABR test revealed that the hearing thresholds shifted to higher frequencies in elderly DBA/2korl mice. In the organ of Corti in the cochlea, damage such as hair cell loss was observed from 13 weeks of age. These hair cells are extremely fragile and cannot regenerate after being damaged [17]. In the case of spiral ganglion, degeneration was observed at 34 weeks of age.

Our study confirmed that this change in hearing function is senile hearing loss, which occurs rapidly in the high-frequency region with age. Taken together, the results obtained here should be useful for senile deafness-related research, understanding the phenotype of laboratory animals, and evaluating the efficacy of new drugs. Of the several mouse models associated with the age-related hearing loss mouse model, the most widely used mouse strain is probably C57BL/6. However, as shown in the results of the previous study [18], C57BL/6 showed age-related hearing loss gradually from 3–5 months, whereas in DBA/2 mice, ABR thresholds are changed more than 20 dB starting from 2 months. Thus, there are more meaningful results about early progressive hearing loss. Although our results focus on the ABR test and histopathology, we tried to compare it sequentially from the younger weeks of 5 weeks old to the older weeks of 13 and 34 weeks of age. These results can be used as reference data for the 5 weeks and 34 weeks of age in the hearing loss study using DBA/2 mice. In addition, DBA/2 mice have shown a tendency to induce auditory seizures in previous studies [19,20,21], and it is known that the *cdh23^ahl^* allele causes age-related hearing loss [12]. These findings suggest that the genetic background of DBA2 mice influences the result of hearing loss with age. Therefore, the DBA2 mouse can be a suitable mouse model for studying the connection between hearing and seizures and the mechanism of hearing loss.

## Figures and Tables

**Figure 1 vetsci-08-00049-f001:**
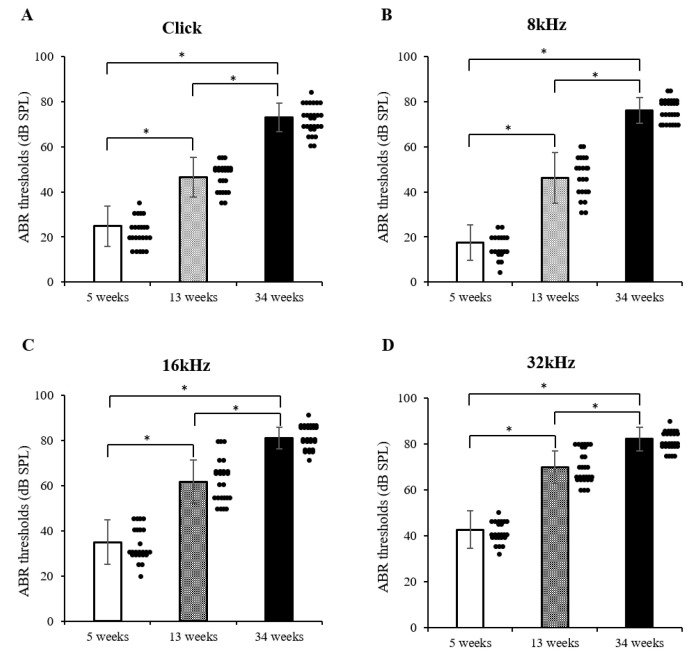
Patterns of hearing threshold in DBA/2korl mice, as revealed by the auditory brainstem response (ABR test): (**A**) transient click, (**B**) 8 kHz, (**C**) 16 kHz, and (**D**) 32 kHz stimuli. Data are presented as mean ± standard error. * Significantly different compared with each group (*n* = 37, * *p* < 0.0001). The data are shown as the mean ± SE.

**Figure 2 vetsci-08-00049-f002:**
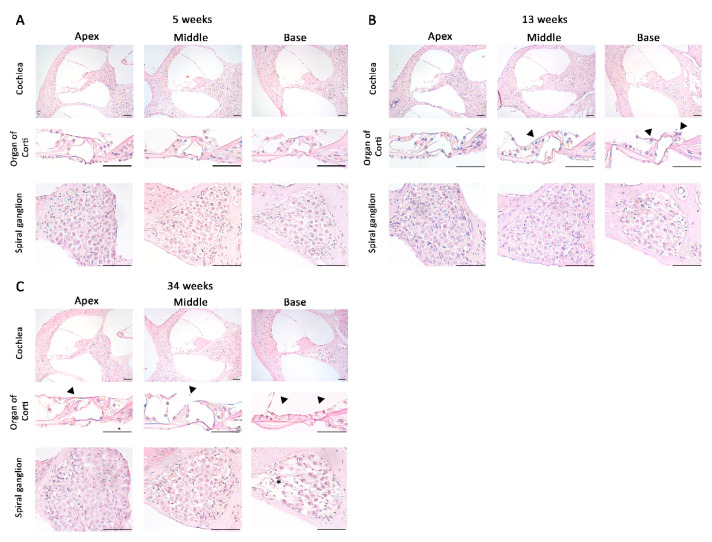
Histopathological analysis of the cochlea in DBA/2korl mice: images of the cochlea of DBA/2 mice at (**A**) 5, (**B**) 13, and (**C**) 34 weeks of age. Black arrowhead: degenerative site on the organ of Corti; asterisk: degenerative site on the spiral ganglion. Scale bars, 100 µm.

## Data Availability

The data presented in this study are available on reasonable request from the corresponding author.

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
