# Peer review of "Phenotype of the Aging-Dependent Spontaneous Onset of Hearing Loss in DBA/2 Mice"

_vetsci, 2021, doi:10.3390/vetsci8030049_

Round 1
Reviewer 1 Report
The paper describes age-related changes in hearing (measured with ABR) and cochlear histology in DBA mice. Although the paper results fill a gap in characterization of this particular mouse strain, there is a very extensive literature on mouse age-related hearing loss in other strains. No context is given here on the particular responses of DBA mice, except a very short mention in the discussion. Please contextualise more, e.g. where would DBA mice be useful in hearing research?
Figure 2 is of such low quality it is impossible to assess the status of most cochlear structures. And figure 1 does not show any raw traces. Error bars are quite small - are they SD or SE?
Moreover, there is no mention anywhere of the number of animals used, and for the histological results data are shown from only one animal for age group. Even for a short communication, a single animal would be too little.
Author Response
Reviewer 1
Comments and Suggestions for Authors
1. The paper describes age-related changes in hearing (measured with ABR) and cochlear histology in DBA mice. Although the paper results fill a gap in characterization of this particular mouse strain, there is a very extensive literature on mouse age-related hearing loss in other strains.
Answer: We thank the reviewer for critical comments on our study. Of the several mouse models associated with the age-related hearing loss mouse model, the most widely used mouse strain is probably C57BL/6. However, as shown in the results of previous study (Ison, Allen et al. 2007), C57BL/6 showed age-related hearing loss gradually from 3-5 months, whereas in DBA/2 mice, ABR thresholds is changed more than 20dB started from 2 months. So, it is more meaningful results about early progressive hearing loss. Although our results focus on the ABR test and histopathology, we tried to compare it sequentially from the younger weeks of 5 weeks old to the old weeks of 13 and 34 weeks of age. These results are considered to be necessary as references in the early stage of 5 weeks of age and the old stage of 34 weeks of age in the group conducting hearing loss studies using DBA/2 mice.
2. No context is given here on the particular responses of DBA mice, except a very short mention in the discussion. Please contextualise more, e.g. where would DBA mice be useful in hearing research?
Answer: As described above, since DBA/2 mice show an early progressive hearing loss, it is easy to use in studies related to progressive hearing loss in the efficacy of therapeutic drugs or in the mechanism of development of hearing loss. In addition, previous reports showing a tendency to cause auditory seizures in DBA/2 mice (Chapman, Croucher et al. 1984, Chapman, De Sarro et al. 1987, Engstrom and Woodbury 1988), so it is a mouse that can be used to study the mechanism and the association between hearing and seizures. According to our results, DBA/2 mice are expected to be suitable for research on early progressive hearing loss, and the presented results of 34 month mice are expected to be used as references in old-age hearing loss studies. So we described in the conclusion section at line 161-163.
3. Figure 2 is of such low quality it is impossible to assess the status of most cochlear structures. And figure 1 does not show any raw traces. Error bars are quite small - are they SD or SE?
Answer: As the reviewer suggested, we improve quality of figure 1. Five weeks age old mouse have normal hearing function, which mouse have normal cochlear structure also. So we used control group for DBA/2 mouse. Error bars in ABR thresholds graph is represented SE. We added the corrected sentence in legend of figure 1 at line 123.
4. Moreover, there is no mention anywhere of the number of animals used, and for the histological results data are shown from only one animal for age group. Even for a short communication, a single animal would be too little.
Answer: In hearing test group, we used 37 male mouse for ABR test and 15 male mouse for histological analysis, not the single mouse. Figure 1 showed representative result of histological analysis. So we added mention of the number of animals in materials and methods section at line 70-71
Reviewer 2 Report
This is a short straightforward manuscript describing the age-dependent hearing loss of DBA/2 mice using ABR recording of the average threshold at three different age stages after birth. The results are in line with the expectation showing the increasing hearing threshold with aging.
The manuscript is short but is not concisely presented. There are sentences that describe ambiguous if not erroneous statement of the observed results.The manuscript also suffers from a redundant description of the same information. Professional editing assistance is highly recommended for the revision. Here are my comments:
- line 25: due to factors intrinsic to these animals. What are the"factors"?
- If there is mean and standard deviation shown in Fig. 1, you should indicate the number of mice used in this experiment.
- Lines 49-52 Incoherent awkward sentence!
- Line 63 rectify --- weeks of age.). There is no beginning parenthesis in this sentence!
- Line 100: four stimuli were "delivered" to the mice not "applied" to the mice!
- Lines 104-105: Wrong!! Fig. 1 shows that the average hearing threshold is <80 dB SPL for click and 8kHz stimuli at 34-week-old mice!!
- Lines 105-106: remove the entire parenthesis! The frequency of these four stimuli has already been described in line 100! Also remove the word "elderly" after 34-week-old. Redundant!
- Fig 1 only shows the threshold of these mice at 5,13,34 weeks of age. The control threshold at 2-3 weeks of age should be shown for comparison! There is no label in the ordinate. readers will have to use their professional knowledge to guess it!
- Line 119: I would think that you "observed" rather than "identified" the normal cochlear structure.
- Lines 142-143> Ambiguous sentence! How can a hearing threshold shift with frequency?It is the elevation of the hearing threshold increases with aging in a greater degree for higher than for lower frequencies.
- Lines 144-145: "Cannot regenerate after being damaged". While it is true that hair cells regeneration is unlikely, this part of the sentence is an overstatement since you did not observe afterward! Unless you cite a reference!
- 12. Line 147: change "the findings of our study" to "Our study".
- Can you elaborate or explain the fact that the high- frequency area (the basal turn) is more susceptible for age-dependent hearing loss than other frequency areas (middle and apical turns)?
Author Response
Reviewer 2
1. line 25: due to factors intrinsic to these animals. What are the"factors"?
Answer: Thank you for your kindly comments. We changed the sentences followed as “DBA/2 mice are a well-known animal model of hearing loss developed due to intrinsic property to these animals.” The corrected sentence was at line 33.
2. If there is mean and standard deviation shown in Fig. 1, you should indicate the number of mice used in this experiment.
Answer: We used totally 37 male mice for ABR test of age-related hearing loss mouse model. We indicated number of mice in legend of figure 1. The corrected sentence was at line 123.
3. Lines 49-52 Incoherent awkward sentence!
Answer: We changed the sentence like “In this case, we selected the age of animal for change of hearing function of senile hearing loss” for coherent paragraph. The corrected sentence was at line 55-56.
4. Line 63 rectify --- weeks of age.). There is no beginning parenthesis in this sentence!
Answer: We removed that parenthesis in sentence.
Before: ~ was conducted at 5, 13, and 34 weeks of age.).
After: ~ was conducted at 5, 13, and 34 weeks of age.
The corrected sentence was at line 72.
5. Line 100: four stimuli were "delivered" to the mice not "applied" to the mice!
Answer: As reviewer’s comment, we changed “applied” to “delivered.
The corrected sentence was at line 109-110.
6. Lines 104-105: Wrong!! Fig. 1 shows that the average hearing threshold is <80 dB SPL for click and 8kHz stimuli at 34-week-old mice!!
Answer: We thank the reviewer for pinpointing the error in this sentence. We fixed it more correctly as below.
Before: In particular, a hearing threshold value of ≥80 dB was recorded for all the four stimuli (transient click and 8, 16, and 32 kHz) in a 34-week-old elderly mouse.
After: In particular, a hearing threshold value was recorded of ≥70 dB in transient click and 8kHz stimuli and ≥80 dB in 16 and 32kHz stimuli in a 34-week-old mouse.
The corrected sentence was at line 114-115.
7. Lines 105-106: remove the entire parenthesis! The frequency of these four stimuli has already been described in line 100! Also remove the word "elderly" after 34-week-old. Redundant!
Answer: We removed all redundant words. We already described in response of number 6.
8. Fig 1 only shows the threshold of these mice at 5,13,34 weeks of age. The control threshold at 2-3 weeks of age should be shown for comparison! There is no label in the ordinate. readers will have to use their professional knowledge to guess it!
Answer: Five weeks age old mouse have normal hearing function, which mouse have normal cochlear structure also. So 5 weeks of age group is control group for hearing function and histopathological analysis. We labelled “ABR thresholds” in the ordinate in figure 1.
9. Line 119: I would think that you "observed" rather than "identified" the normal cochlear structure.
Answer: We changed the sentences as reviewer’s suggestion.
Before: At 5 weeks of age, a normal histopathological structure was identified (Figure 2A).
After: At 5 weeks of age, a normal histopathological structure was observed (Figure 2A).
The corrected sentence was at line 129.
10. Lines 142-143> Ambiguous sentence! How can a hearing threshold shift with frequency? It is the elevation of the hearing threshold increases with aging in a greater degree for higher than for lower frequencies.
Answer: We changed the sentences as reviewer’s suggestion.
Before: In the present study, the ABR test revealed that the hearing threshold shifted to higher frequencies with an increasing age in DBA/2korl mice.
After: In the present study, the ABR test revealed that the hearing threshold more shifted in high frequencies in elderly DBA/2korl mice.
The corrected sentence was at line 152-153.
11. Lines 144-145: "Cannot regenerate after being damaged". While it is true that hair cells regeneration is unlikely, this part of the sentence is an overstatement since you did not observe afterward! Unless you cite a reference!
Answer: It is widely known that, as suggested by the reviewer, damage to hair cells is not regenerated (Smith, Groves et al. 2016). So we added a reference about hair cell death and regeneration in the sentence.
The corrected sentence was at line 155 and 202-203.
12. Line 147: change "the findings of our study" to "Our study".
Answer: We have changed “the findings of our study” to “Our study”.
The corrected sentence was at line 157.
13. Can you elaborate or explain the fact that the high- frequency area (the basal turn) is more susceptible for age-dependent hearing loss than other frequency areas (middle and apical turns)?
Answer: There is no clear mechanism for worsening high-frequency hearing in senile hearing loss. However, the results of several studies with age-related hearing loss have reported that hearing at almost high frequencies falls first (White, Burgess et al. 2000, Stamataki, Francis et al. 2006, Ison, Allen et al. 2007, Kim, Baek et al. 2019). These phenomena are expected to be related to ROS, noise, etc., which are the main causes of age-related hearing loss (Smith, Groves et al. 2016).
References
Chapman, A. G., M. J. Croucher and B. S. Meldrum (1984). "Evaluation of anticonvulsant drugs in DBA/2 mice with sound-induced seizures." Arzneimittelforschung 34(10): 1261-1264.
Chapman, A. G., G. B. De Sarro, M. Premachandra and B. S. Meldrum (1987). "Bidirectional effects of beta-carbolines in reflex epilepsy." Brain Res Bull 19(3): 337-346.
Engstrom, F. L. and D. M. Woodbury (1988). "Seizure susceptibility in DBA and C57 mice: the effects of various convulsants." Epilepsia 29(4): 389-395.
Ison, J. R., P. D. Allen and W. E. O'Neill (2007). "Age-related hearing loss in C57BL/6J mice has both frequency-specific and non-frequency-specific components that produce a hyperacusis-like exaggeration of the acoustic startle reflex." J Assoc Res Otolaryngol 8(4): 539-550.
Kim, Y. R., J. I. Baek, S. H. Kim, M. A. Kim, B. Lee, N. Ryu, K. H. Kim, D. G. Choi, H. M. Kim, M. P. Murphy, G. Macpherson, Y. S. Choo, J. Bok, K. Y. Lee, J. W. Park and U. K. Kim (2019). "Therapeutic potential of the mitochondria-targeted antioxidant MitoQ in mitochondrial-ROS induced sensorineural hearing loss caused by Idh2 deficiency." Redox Biol 20: 544-555.
Smith, M. E., A. K. Groves and A. B. Coffin (2016). "Editorial: Sensory Hair Cell Death and Regeneration." Front Cell Neurosci 10: 208.
Stamataki, S., H. W. Francis, M. Lehar, B. J. May and D. K. Ryugo (2006). "Synaptic alterations at inner hair cells precede spiral ganglion cell loss in aging C57BL/6J mice." Hear Res 221(1-2): 104-118.
White, J. A., B. J. Burgess, R. D. Hall and J. B. Nadol (2000). "Pattern of degeneration of the spiral ganglion cell and its processes in the C57BL/6J mouse." Hear Res 141(1-2): 12-18.
Round 2
Reviewer 1 Report
- The paper describes age-related changes in hearing (measured with ABR) and cochlear histology in DBA mice. Although the paper results fill a gap in characterization of this particular mouse strain, there is a very extensive literature on mouse age-related hearing loss in other strains.
Answer: We thank the reviewer for critical comments on our study. Of the several mouse models associated with the age-related hearing loss mouse model, the most widely used mouse strain is probably C57BL/6. However, as shown in the results of previous study (Ison, Allen et al. 2007), C57BL/6 showed age-related hearing loss gradually from 3-5 months, whereas in DBA/2 mice, ABR thresholds is changed more than 20dB started from 2 months. So, it is more meaningful results about early progressive hearing loss. Although our results focus on the ABR test and histopathology, we tried to compare it sequentially from the younger weeks of 5 weeks old to the old weeks of 13 and 34 weeks of age. These results are considered to be necessary as references in the early stage of 5 weeks of age and the old stage of 34 weeks of age in the group conducting hearing loss studies using DBA/2 mice. - REPLY: Please add information TO THE PAPER, not to this reply, otherwise this is a bit useless.
2. No context is given here on the particular responses of DBA mice, except a very short mention in the discussion. Please contextualise more, e.g. where would DBA mice be useful in hearing research?
Answer: As described above, since DBA/2 mice show an early progressive hearing loss, it is easy to use in studies related to progressive hearing loss in the efficacy of therapeutic drugs or in the mechanism of development of hearing loss. In addition, previous reports showing a tendency to cause auditory seizures in DBA/2 mice (Chapman, Croucher et al. 1984, Chapman, De Sarro et al. 1987, Engstrom and Woodbury 1988), so it is a mouse that can be used to study the mechanism and the association between hearing and seizures. According to our results, DBA/2 mice are expected to be suitable for research on early progressive hearing loss, and the presented results of 34 month mice are expected to be used as references in old-age hearing loss studies. So we described in the conclusion section at line 161-163.
REPLY: same as previous point, this discussion needs to be added to the paper. Moreover, there are many types of hearing loss, and given that the genetic background of DBA mice has been studied in several papers, it would be important to discuss at least the known vulnerabilities of this strain. And right now there is none of that in the paper, except a too vague definition of "early progressive hearing loss".
3. Figure 2 is of such low quality it is impossible to assess the status of most cochlear structures. And figure 1 does not show any raw traces. Error bars are quite small - are they SD or SE?
Answer: As the reviewer suggested, we improve quality of figure 1.
REPLY: that was figure 2. Can you upload a high res version?
Five weeks age old mouse have normal hearing function, which mouse have normal cochlear structure also. So we used control group for DBA/2 mouse.
REPLY: I am not sure what this means. What is "control" in an aging study?
Error bars in ABR thresholds graph is represented SE. We added the corrected sentence in legend of figure 1 at line 123.
With 37 animals, histograms are not very informative about the distribution of hearing loss. Please add points indicating single experiments, or transform histograms in violin plots, in order to give better information on population dispersion.
4. Moreover, there is no mention anywhere of the number of animals used, and for the histological results data are shown from only one animal for age group. Even for a short communication, a single animal would be too little.
Answer: In hearing test group, we used 37 male mouse for ABR test and 15 male mouse for histological analysis, not the single mouse. Figure 1 showed representative result of histological analysis. So we added mention of the number of animals in materials and methods section at line 70-71
What was the rationale behind such a high number of animals (especially for ABR) given that the expected effect was presumably quite large, given the data already available in the literature?
Author Response
Comments and Suggestions for Authors
1. The paper describes age-related changes in hearing (measured with ABR) and cochlear histology in DBA mice. Although the paper results fill a gap in characterization of this particular mouse strain, there is a very extensive literature on mouse age-related hearing loss in other strains.
Answer: We thank the reviewer for critical comments on our study. Of the several mouse models associated with the age-related hearing loss mouse model, the most widely used mouse strain is probably C57BL/6. However, as shown in the results of previous study (Ison, Allen et al. 2007), C57BL/6 showed age-related hearing loss gradually from 3-5 months, whereas in DBA/2 mice, ABR thresholds is changed more than 20dB started from 2 months. So, it is more meaningful results about early progressive hearing loss. Although our results focus on the ABR test and histopathology, we tried to compare it sequentially from the younger weeks of 5 weeks old to the old weeks of 13 and 34 weeks of age. These results are considered to be necessary as references in the early stage of 5 weeks of age and the old stage of 34 weeks of age in the group conducting hearing loss studies using DBA/2 mice.
REPLY: Please add information TO THE PAPER, not to this reply, otherwise this is a bit useless.
Answer: As reviewer’s suggestion, we added the paragraph followed as “Of the several mouse models associated with the age-related hearing loss mouse model, the most widely used mouse strain is probably C57BL/6. However, as shown in the results of previous study (18), C57BL/6 showed age-related hearing loss gradually from 3-5 months, whereas in DBA/2 mice, ABR thresholds is changed more than 20dB started from 2 months. So, it is more meaningful results about early progressive hearing loss. Although our results focus on the ABR test and histopathology, we tried to compare it sequentially from the younger weeks of 5 weeks old to the old weeks of 13 and 34 weeks of age. These results are considered to be necessary as references in the early stage of 5 weeks of age and the old stage of 34 weeks of age in the group conducting hearing loss studies using DBA/2 mice.“ at line 155-164.
2. No context is given here on the particular responses of DBA mice, except a very short mention in the discussion. Please contextualise more, e.g. where would DBA mice be useful in hearing research?
Answer: As described above, since DBA/2 mice show an early progressive hearing loss, it is easy to use in studies related to progressive hearing loss in the efficacy of therapeutic drugs or in the mechanism of development of hearing loss. In addition, previous reports showing a tendency to cause auditory seizures in DBA/2 mice (Chapman, Croucher et al. 1984, Chapman, De Sarro et al. 1987, Engstrom and Woodbury 1988), so it is a mouse that can be used to study the mechanism and the association between hearing and seizures. According to our results, DBA/2 mice are expected to be suitable for research on early progressive hearing loss, and the presented results of 34 month mice are expected to be used as references in old-age hearing loss studies. So we described in the conclusion section at line 161-163.
REPLY: same as previous point, this discussion needs to be added to the paper. Moreover, there are many types of hearing loss, and given that the genetic background of DBA mice has been studied in several papers, it would be important to discuss at least the known vulnerabilities of this strain. And right now there is none of that in the paper, except a too vague definition of "early progressive hearing loss".
Answer: We thank the reviewer for critical comments on our study. We have already added this to the paper. However, as the reviewer said, in the case of DBA2, age-related hearing loss has been shown to occur due to the cdh23ahl allele related to age-related hearing loss. It can be seen that the genetic background of these DBA2 mice affects our results of hearing loss with age. The above was revised followed as “age-related hearing loss has been shown to occur due to the cdh23ahl allele related to age-related hearing loss (12). It can be seen that the genetic background of these DBA2 mice affects our results of hearing loss with age.” in the conclusion section at line 166-168.
3. Figure 2 is of such low quality it is impossible to assess the status of most cochlear structures. And figure 1 does not show any raw traces. Error bars are quite small - are they SD or SE?
Answer: As the reviewer suggested, we improve quality of figure 1.
REPLY: that was figure 2. Can you upload a high res version?
Answer: Now the figure is a high quality figure of 300dpi. When viewed at a magnified view, it is considered to be at a level where the cochlea structure can be sufficiently confirmed, and this image quality figure that we can make at the maximum.
4. Five weeks age old mouse have normal hearing function, which mouse have normal cochlear structure also. So we used control group for DBA/2 mouse.
REPLY: I am not sure what this means. What is "control" in an aging study?
Answer: In age-related hearing loss experiments, hearing changes with age should be compared. Therefore, it is important to observe how early-stage individuals exhibit their hearing when they are older. Therefore, we took the hearing function of the 5 weeks old mouse as a reference point to compare how much of the hearing loss of the 5 weeks old mouse, which is a child of DBA2, at 34 weeks old. Hearing in 5 weeks old mice showed little difference between individuals, and hearing was generally measured in the normal hearing function threshold range (5-40 dB SPL) (Zheng Q. Y. et al., 1999, Hear Res).
5. Error bars in ABR thresholds graph is represented SE. We added the corrected sentence in legend of figure 1 at line 123.
REPLY: With 37 animals, histograms are not very informative about the distribution of hearing loss. Please add points indicating single experiments, or transform histograms in violin plots, in order to give better information on population dispersion.
Answer: As reviewer’s suggestion, we added raw traces indicating single experiments to the histogram.
6. Moreover, there is no mention anywhere of the number of animals used, and for the histological results data are shown from only one animal for age group. Even for a short communication, a single animal would be too little.
Answer: In hearing test group, we used 37 male mouse for ABR test and 15 male mouse for histological analysis, not the single mouse. Figure 1 showed representative result of histological analysis. So we added mention of the number of animals in materials and methods section at line 70-71
REPLY: What was the rationale behind such a high number of animals (especially for ABR) given that the expected effect was presumably quite large, given the data already available in the literature?
Answer: In the case of age-related hearing loss, since there may be a lot of differences between individuals, we tried to check whether a certain age-related hearing loss appeared by using a large number of mice. As a result, we could confirm that most of the mice have similar hearing function.
References
Zheng Q.Y. et al. Assessment of hearing in 80 inbred strains of mice by ABR threshold analyses. Hear Res. 1999;130(1-2):94-107
